# Identification of the Potential Critical Slip Surface for Fractured Rock Slope Using the Floyd Algorithm

**Shengyuan Song** [1], **Mingyu Zhao** [1], **Chun Zhu** [1,2,3,*], **Fengyan Wang** [4], **Chen Cao** [1], **Haojie Li** [1] and **Muye Ma** [1]

1. College of Construction Engineering, Jilin University, Changchun 130026, China; songshengyuan@jlu.edu.cn (S.S.); myzhao20@mails.jlu.edu.cn (M.Z.); ccao@jlu.edu.cn (C.C.); lihj21@mails.jlu.edu.cn (H.L.); mamy21@mails.jlu.edu.cn (M.M.)
2. School of Earth Sciences and Engineering, Hohai University, Nanjing 210098, China
3. Shaanxi Key Laboratory of Advanced Stimulation Technology for Oil & Gas Reservoirs, College of Petroleum Engineering, Xi'an Shiyou University, Xi'an 710065, China
4. College of Geo-Exploration Science and Technology, Jilin University, Changchun 130026, China; wangfy@jlu.edu.cn
* Correspondence: zhu.chun@hhu.edu.cn

**Abstract:** A rock slope can be characterized by tens of persistent discontinuities. A slope can be massive. The slip surface of the slope is usually easier to expand along with the discontinuities because the shear strength of the discontinuities is substantially lower than that of the rock blocks. Based on this idea, this paper takes a jointed rock slope in Hengqin Island, Zhuhai as an example, and establishes a three-dimensional (3D) model of the studied slope by digital close-range photogrammetry to rapidly interpret 222 fracture parameters. Meanwhile, a new Floyd algorithm for finding the shortest path is developed to realize the critical slip surface identification of the studied slope. Within the 3D fracture network model created using the Monte Carlo method, a sequence of cross-sections is placed. These cross-sections containing fractures are used to search for the shortest paths between the designated shear entrances and exits. For anyone combination of entry point and exit point, the shortest paths corresponding to different cross-sections are different and cluttered. For the sake of safety and convenience, these shortest paths are simplified as a circular arc that is regarded as a potential slip surface. The fracture frequency is used to determine the probability of sliding along a prospective critical slip surface. The potential slip surface through the entrance point (0, 80) and exit point (120, 0) is identified as the final critical slip surface of the slope due to the maximum fracture frequency.

**Keywords:** rock slope; digital close-range photogrammetry; critical slip surface; Floyd algorithm; fracture network

## 1. Introduction

Slope instability will destroy buildings, block roads, and seriously threaten human life and property safety [1–3]. Therefore, slope stability analysis is an important part of geological engineering and geotechnical engineering research to prevent slope destruction. Slope stability analysis relies heavily on identifying the crucial slip surface and calculating the safety factor [4–6]. However, only the critical slip surface can be accurately determined, upon which the calculation of the safety factor is valid. As a result, the fundamental challenge in slope stability analysis is determining the critical slip surface.

Critical slip surface studies on soil slopes are more mature, and the slip surface is typically viewed as circular [7]. Some mathematical programming methods and intelligent algorithms, such as the artificial fish swarms algorithm, the ant colony algorithm, genetic algorithms, the imperialistic competitive algorithm, and the black hole algorithm, have been used in recent years to locate the crucial slip surface [8–12]. Although these methods produce good results in identifying the critical slip surface of a soil slope, they cannot be

directly applied to identifying the critical slip surface created by fracture propagation on a rock slope [13].

For rock slopes, the studies of critical slip surfaces are relatively rare, and the slip surface is ordinarily linear or folds linear that is composed of multiple fractures [14]. Einstein et al. suggested the effect of fracture persistence on rock slope stability and developed a method for connecting slope stability and hence persistence to fracture geometry and spatial variability [15]. Zhu et al. established a method for calculating the critical slip surface of a rock slope according to the limit equilibrium method and optimization theory [16]. However, this method only considers the fractures when tracking the exit points of the slip surface. In addition, due to the complexity of the rock mass structure, the influence of the dip angle discontinuity on slope stability cannot be fully considered by this method. Zhang et al. proposed a method for determining the slip surface based on the frequency of the fracture orientation [17]. In this study, the stochastic dynamics method is improved, but because only the orientation of fracture is considered, the influence of parameters such as the trace length is not analyzed. Therefore, this method can determine the orientation of the slip surface, however not the specific location of the slip surface. This method is only suitable for determining the critical slip surface of fractured rock mass with a similar trace length. Zheng et al. proposed a new stress-based search technique for obtaining the critical slip surface and safety factor directly [18]. This method combines the numerical manifold method and graph theory, reduces the iterative computation, and eliminates the rigid body assumption in the limit equilibrium method. Although this method can consider the influence of parallel regular fractures, it is not consistent with the random distribution of fractures in the field. Liu et al. [19] proposed a new method to determine the safety factor and the corresponding critical slip surface of a seismic excited substratum rock slope. The method combines shear strength reduction technology and 2D discrete element program UDEC, which is not affected by subjective factors and can directly obtain results. However, only layered rock slopes are considered in this study, and other types of rock slopes are not involved. Xu et al. established a new model to search the critical slip surface according to the genetic algorithm [20]. However, this study indirectly considers the effect of the fractures on slope stability from the perspective of the stress field.

The slip surface of a rock slope with a complex structure is formed by the extension and connection of preexisting discontinuities. To put it another way, determining the crucial slip surface should take into account all of the pre-existing fracture data, including its position, orientation, and size. As a result, a method for finding arbitrarily shaped slip surfaces that are suited for rock slopes is required. In this paper, close-range photogrammetry technology is used to quickly obtain random discontinuities of rock slopes to establish the fracture network model, and the Floyd algorithm is used to find the critical slip surface of a complicated rock slope.

## 2. Materials and Methods

### 2.1. Study Area

The Nanshanju slope on Hengqin Island is used to determine the critical slip surface in this study. The Hengqin Island is suited in the southeast of Zhuhai, Guangdong Province, China, and is adjacent to Macao, China (Figure 1).

The Hengqin Island has two mountains, which are named the Small Hengqin Mountain and the Large Hengqin Mountain, respectively. Due to artificial quarrying, the Nanshanju slope has formed in the east of the Small Hengqin Mountain. The Nanshanju slope is more than 600 m long and nearly 80 m high. The dip direction of the slope is approximately 87°.

The Nanshanju slope is classified into five zones based on differences in slope surface orientation: slope 1, slope 2, slope 3, slope 4, and slope 5. Slope 1 is randomly selected as the paper's principal study object (Figure 2). According to the field investigation, slope 1 has the highest height, the largest slope area, and the steepest slope, and is more prone to sliding and failure. It is a more typical slope for study.

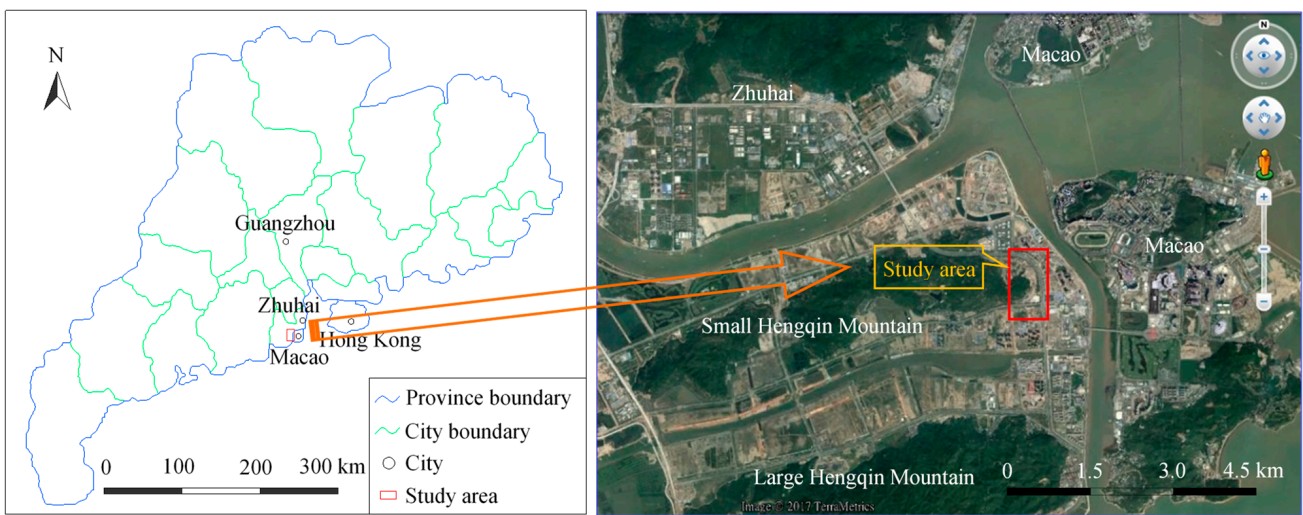

**Figure 1.** Location and topography of the study area.

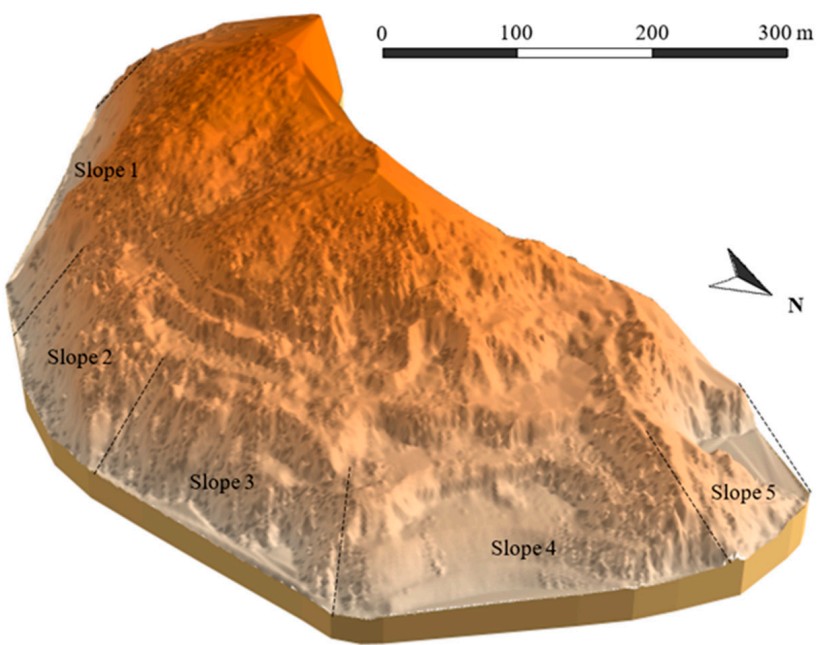

**Figure 2.** Three-dimensional model of the Nanshanju slope.

The dip direction of slope 1 is 139°. The excavation marks of the slope surface are remarkable. The main lithology of the slope body consists of medium-coarse granite from the Yanshan period. The fresh surface is flesh red and the weathered surface is light grey to dark grey (Figure 3).

The weathering degree of the slope is quite different and is mainly weak to strong. The large number of developed discontinuities reduces the mechanical properties of the rock mass. Dangerous rock bodies are visible on the slope surface. It is urgent to collect a large number of random discontinuities for slope stability analysis.

### 2.2. Collecting Discontinuity by Digital Close-Range Photogrammetry

As the slope in the study area is very high and steep, only the discontinuity at the bottom of the slope can be measured manually, so the digital close-range photogrammetry technology is used to obtain the whole slope discontinuity.

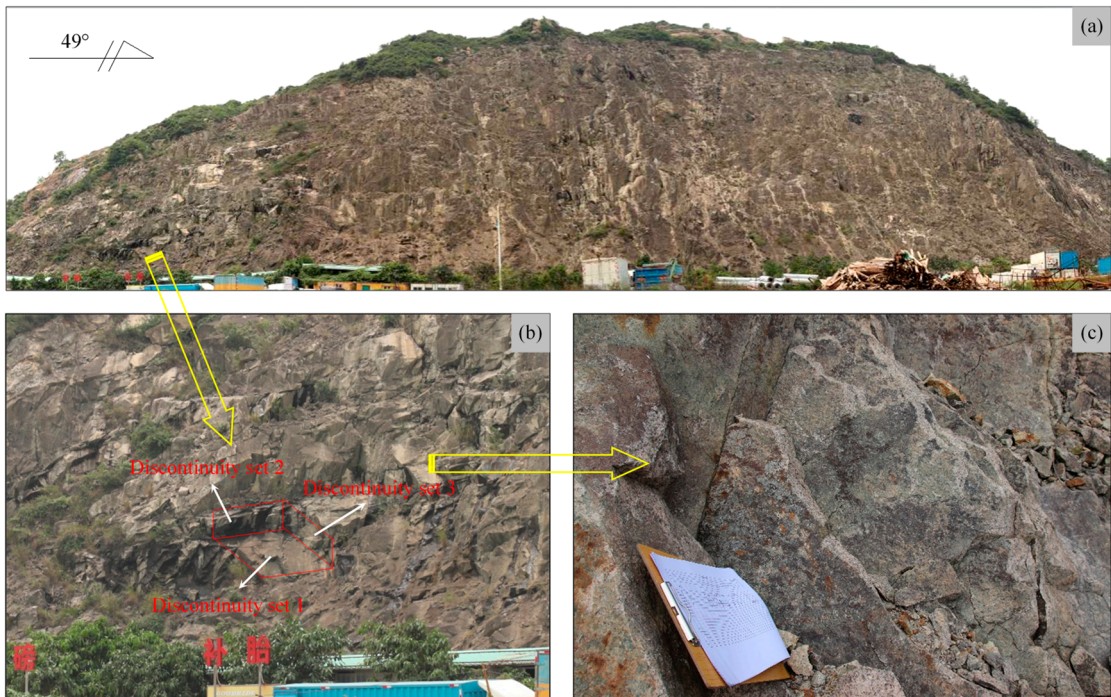

**Figure 3.** Fractures and dangerous rock bodies developed on the slope surface; (**a**) Overall shape of slope; (**b**) Fracture development; (**c**) Dangerous block.

The first step of close-range photogrammetry is a hierarchical control measurement. The GPS-RTK technique is used in the first-level ground control measurement to acquire the 3D coordinates of ground control points in the chosen coordinate system. Then stable rock blocks are selected or concrete stakes are made with paint spraying as ground control point markers in open view as far as possible. It is also important to ensure that the control points are in view of each other to facilitate the implementation of the slope control measurement.

Based on the results of the first-level control measurement, the secondary slope control measurement uses the spatial polar coordinate measurement method to obtain the 3D coordinates of the slope control points under the specified coordinate system through the prism-free mode of a total station. Distinctive color spot corner points on the rock as slope control point markers are selected. The position of the edge of the rock mass should not be chosen to avoid errors caused by the laser reflections of the total station.

After the control measurement, the slope is photographed next. Usually, a non-metric camera that has been calibrated is used to take stereo images of the slope by means of orthographic photography (Figure 4). The photography baseline should be set up parallel to the slope strike, and the length of the photography baseline should be 1/5~1/10 of the photograph distance. At the same time, the camera's main optical axis should be parallel to each other and perpendicular to the photographic baseline when photographing twice. The image overlap of the image pair is greater than 80%, and at least 6 slope control points are included on each image.

Image correction is required after digital photography. Then, the relationship between image point coordinates and object space coordinates is established by the mathematical model of close-range photogrammetry—direct linear transformation algorithm [21]:

$$\begin{cases} x - x_0 + \Delta x + \dfrac{l_1 X + l_2 Y + l_3 Z + l_4}{l_9 X + l_{10} Z + l_{11} + 1} = 0 \\ y - y_0 + \Delta y + \dfrac{l_5 X + l_6 Y + l_7 Z + l_8}{l_9 X + l_{10} Z + l_{11} + 1} = 0 \end{cases} \tag{1}$$

where $l_i$ (i = 1, 2, ..., 11) is the direct linear transformation coefficient; $(x_0, y_0)$ is the image principal point coordinates; and $\Delta x$ and $\Delta y$ are the nonlinear correction numbers of the image point coordinates, which can be expressed as:

$$\begin{cases} \Delta x = (x - x_0)(K_1 r^2 + K_2 r^4) + P_1[r^2 + 2(x - x_0)^2] + 2P_2(x - x_0)(y - y_0) \\ \Delta y = (y - y_0)(K_1 r^2 + K_2 r^4) + P_1[r^2 + 2(y - y_0)^2] + 2P_2(x - x_0)(y - y_0) \end{cases} \quad (2)$$

where $K_1$, $K_2$ are radial aberration coefficients; $P_1$, $P_2$ are tangential aberration coefficients; and $r$ is the image point vectorial diameter, which can be expressed as:

$$r = \sqrt{(x - x_0)^2 + (y - y_0)^2}. \quad (3)$$

VirtuoZo is a highly intelligent digital photogrammetry system that can quickly orient digital images and generate 3D models. This paper uses a VirtuoZo system to identify fractures and measure the feature points. The Canon 5D II digital camera is used in this study, so it is necessary to set the image type as a non-metric camera during operation. After obtaining the 3D model of the studied slope, the trace length and orientation of the fracture can be interpreted by measuring some feature points.

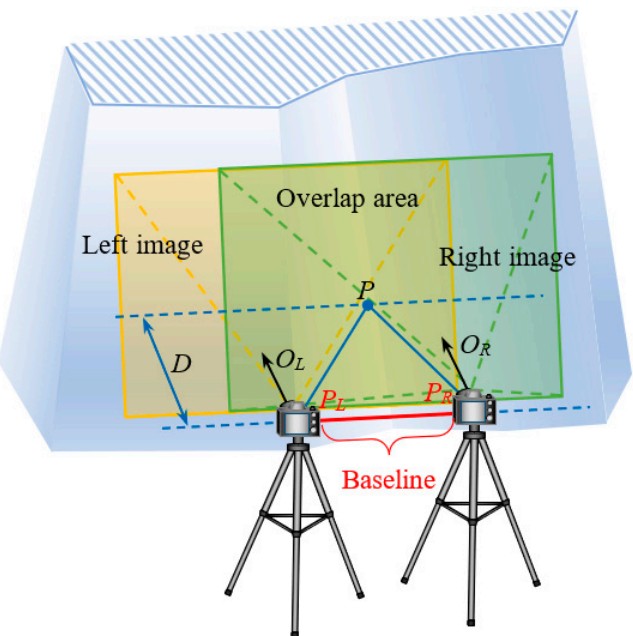

**Figure 4.** Schematic diagram of digital close-range photogrammetry.

In this paper, the longest line segment in the visible range of the fracture is defined as the trace length of the fracture. Therefore, the trace endpoint coordinates can be measured directly on the 3D model, and the trace length $d$ can be solved by Formula (4):

$$d = \sqrt{(x_1 - x_2)^2 + (y_1 - y_2)^2 + (z_1 - z_2)^2}. \quad (4)$$

In this paper, the dip direction and dip angle are used to represent the orientation of a fracture. Suppose the equation of a fracture is $Z = Ax + By + C$, and its normal vector is t $(-A, -B, 1)$. $n$ $(n \geq 3)$ feature points are measured on the fracture, then $A$, $B$, and $C$ can be solved by the least square method:

$$\begin{bmatrix} A \\ B \\ C \end{bmatrix} = \left( \begin{bmatrix} X_1 & Y_1 & 1 \\ X_2 & Y_2 & 1 \\ \vdots & \vdots & \vdots \\ X_n & Y_n & 1 \end{bmatrix}^T \begin{bmatrix} X_1 & Y_1 & 1 \\ X_2 & Y_2 & 1 \\ \vdots & \vdots & \vdots \\ X_n & Y_n & 1 \end{bmatrix} \right)^{-1} \begin{bmatrix} X_1 & Y_1 & 1 \\ X_2 & Y_2 & 1 \\ \vdots & \vdots & \vdots \\ X_n & Y_n & 1 \end{bmatrix}^T \begin{bmatrix} Z_1 \\ Z_2 \\ \vdots \\ Z_n \end{bmatrix}. \quad (5)$$

Then the dip direction $\beta$ and dip angle $\alpha$ can be obtained:

When $A = 0$:

$$
\begin{cases}
\alpha = |\arctan(B)| \\
\beta = \begin{cases}
\pi/2, B < 0 \\
3\pi/2, B > 0 \\
\forall, B = 0
\end{cases}
\end{cases}
\tag{6}
$$

When $A \neq 0$:

$$
\begin{cases}
\alpha = \left|\arctan(\sqrt{A^2 + B^2})\right| \\
\beta = \begin{cases}
\arctan(B/A), A < 0, B \leq 0 \\
\arctan(B/A) + 2\pi, A < 0, B > 0 \\
\arctan(B/A) + \pi, A > 0
\end{cases}
\end{cases}
\tag{7}
$$

During operation, the area enclosed by feature points should be as large as possible to ensure that the polygon enclosed by feature points can represent the whole fracture. If three feature points are selected, the interior angles should be uniform, not less than 30° or not more than 150°.

By digital close-range photogrammetry, 222 fractures are collected from the surface of slope 1. By comparing with the field measurement of discontinuity orientation, the average error of dip direction and dip angle obtained by digital close-range photogrammetry interpretation is 3° and 2°, respectively, which meets the requirement of accuracy. These fractures are used to establish the discrete fracture model.

### 2.3. Fracture Generation in the Cross-Section of the Slope

To analyze the critical slip surface of slope 1 in the 2D plane, fractures in the cross-section perpendicular to the slope surface need to be generated. First, the fracture information obtained from the slope surface is used to create a 3D fracture network model. Second, the 3D fracture network model is used to extract fracture information in the cross-section. The specific steps are as follows [22].

#### 2.3.1. Analysis of Fracture Orientation

Based on the fracture orientation data, an improved fuzzy C-means (FCM) method introduced by Song et al. [23] and is utilized to identify the fracture sets. Rock masses are characterized by three discontinuity sets, as shown in Figure 5.

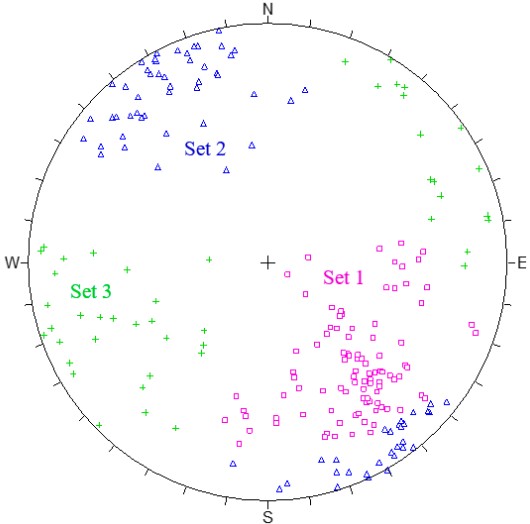

**Figure 5.** Grouping of the fractures according to the orientations.

Subsequently, the weighting function $W_i$ proposed by Kulatilake & Wu [24] is used to correct the orientation bias for each fracture set. In a rectangular sampling window of width $w$ and height $h$, $W_i$ is defined as:

$$W_i = \left[ \left( \cos^2 \theta_i \sin^2 \beta_i + \cos^2 \beta_i \right)^{0.5} + \frac{\pi d_i}{4h} \cos \beta_i \sin \theta_i + \frac{\pi d_i}{4w} \cos \theta_i \right]^{-1} \quad (8)$$

where $\theta_i$ and $d_i$ indicate the dip angle and the diameter of the $i$th fracture, respectively. The angle between the window plane's strike and the dip direction of the $i$th fracture is represented by $\beta_i$.

In addition, the clustering degree for the fracture set is determined by calculating the Fisher constant. The following formula can be used to compute the Fisher constant [25]:

$$K = \frac{M}{M - \sum\limits_{i=1}^{M} |r_i|} \quad (9)$$

where $K$ expresses the Fisher constant. $r_i$ indicates the unit normal vector for the $i$th fracture. $M$ represents the fracture number. The Fisher constant for each fracture set is shown in Table 1.

**Table 1.** The parameters of fractures used to generate the 3D network model.

| Fracture Set | Fracture Number | Trace Length | | | Fracture Diameter | | | Density (m$^{-3}$) | K |
|---|---|---|---|---|---|---|---|---|---|
| | | Mean (m) | Std. (m) | Distribution Type | Mean (m) | Std. (m) | Distribution Type | | |
| 1 | 99 | 12.97 | 14.87 | Gamma | 16.98 | 4.73 | Gamma | 0.000544 | 12.97 |
| 2 | 77 | 3.56 | 5.02 | Gamma | 5.52 | 1.33 | Gamma | 0.001035 | 13.78 |
| 3 | 46 | 6.05 | 7.75 | Gamma | 7.58 | 2.94 | Gamma | 0.000422 | 5.51 |

### 2.3.2. Determination of Trace Length, Diameter, and Density

The observed trace lengths are shorter than the true trace lengths due to the sample window's limitations. The mean of the observed trace lengths is corrected using the method suggested by Zhang et al. [26]:

$$\mu_m = \frac{C_m w h}{\sum_1^m l_i \int_0^{90} \sin \alpha_i g(\alpha_i) d\alpha_i - C_m [w \int_0^{90} \sin \theta_i g(\theta_i) d\theta_i + h \int_0^{90} \cos \theta_i g(\theta_i) d\theta_i]} \quad (10)$$

where $\mu_m$ is the mean trace length after correction, and $m$ represents the number of scanlines. $l_i$ denotes the length of the scanline $L_i$. $a_i$ indicates the angle between the scanline $L_i$ and the fracture trace. $g(\alpha_i)$ and $g(\theta_i)$ are the probability density functions of $\alpha_i$ and $\theta_i$, respectively. $C_m = \sum_1^m R_{L_i}$ is the percentage of the expected number of traces intersecting with all scanlines. In 3D space, the fracture is generally considered a disc. A numerical solution scheme established by Kulatilake and Wu [27] is utilized to acquire the probability distribution of the disc diameter according to the probability density of the trace length.

Based on the surveyed fracture information, the 2D fracture traces can be drafted by computer technology. In the 2D fracture network, scanlines with different directions are set and used to measure the fracture spacing. Then, the line density $\lambda_j^1$ of the fractures can be obtained for each fracture set. Based on these results, the volume density $\lambda_j^V$ of the fractures can be calculated by [28]:

$$\lambda_j^V = \frac{4 \lambda_j^1}{\pi E(D^2)} \quad (11)$$

where $E(D^2)$ denotes the second moment of fracture diameter $D$.

The trace length, diameter, and density of each fracture set are determined using the procedures described above, as shown in Table 1.

### 2.3.3. Monte Carlo Simulation and Model Verification

Based on the scale of the rock slope, a 3D fracture network model with dimensions of 200 m × 120 m × 80 m is designed. The fracture number is determined by the fracture density and model size. The Monte Carlo approach is then used to simulate probability distribution models of fracture orientation, diameter, and location as specified by the fracture space. The empirical probability distribution is used in this work to define the orientation of each fracture set. Each fracture set's diameter follows a gamma distribution. Each fracture set's placement follows a consistent distribution. Each model is simulated 5 times. Through the combination of the three models, a total of 125 3D fracture network models are established.

A sample window with the same size and orientation as the field survey is arranged in the 3D fracture network model to find a trustworthy fracture network model. The fracture parameters gathered from the artificial sample window are then compared to the field data. The ideal model is a 3D fracture model with a high degree of resemblance to the field data. To determine if the field data and simulated data are in close agreement, two criteria are used: the relative difference percentage (*RD*%) and the coefficient of variation (*COV*) [29,30]:

$$RD\% = \frac{Average \quad simulated \quad value - Field \quad value}{Field \quad value} \times 100\% \tag{12}$$

$$COV = \frac{Std.}{Mean} \times 100\%. \tag{13}$$

The compared results and optimal model are shown in Table 2 and Figure 6.

**Table 2.** The validation of the optimal model by comparing the surveyed data and simulated data.

| Fracture Set | Data Type | Fracture Number | Mean Orientation (°) | | Mean Trace Length (m) | | Trace Type | | | K |
|---|---|---|---|---|---|---|---|---|---|---|
| | | | Dip Direction | Dip Angle | Observed | Corrected | $R_0$ | $R_1$ | $R_2$ | |
| 1 | Surveyed | 99 | 140.8 | 51.2 | 12.97 | 16.05 | 0.00 | 0.07 | 0.93 | 12.97 |
| | Simulated | 95 | 141.3 | 48.4 | 12.33 | 12.43 | 0.00 | 0.28 | 0.72 | 15.60 |
| 2 | Surveyed | 77 | 150.8 | 85.2 | 3.56 | 5.04 | 0.00 | 0.04 | 0.96 | 13.78 |
| | Simulated | 75 | 148.0 | 84.8 | 4.54 | 5.45 | 0.00 | 0.15 | 0.85 | 14.70 |
| 3 | Surveyed | 46 | 246.9 | 81.8 | 6.05 | 8.02 | 0.00 | 0.07 | 0.93 | 5.51 |
| | Simulated | 44 | 249.5 | 77.8 | 5.94 | 8.47 | 0.00 | 0.19 | 0.81 | 5.70 |

### 2.3.4. Fracture Extraction in the Cross-Section

The slip surface determined on the 3D scale is more reasonable and accurate. The cracks formed within the rock slope, on the other hand, are exceedingly complicated, making it impossible to search the slide surface on a 3D scale. Therefore, the slip surface is first determined from the 2D scale.

To facilitate the search for the slip surface of the slope, a series of cross-sections are set within the 3D fracture network model. These cross-sections are perpendicular to the *x*-axis and arranged at *x* = 10 m, 20 m, . . . , 190 m (Figure 7).

Due to the intersection of the section with the fracture in the model, a 2D fracture network is generated in each cross-section. By looking for the shortest path, 19 cross-sections with a 2D fracture network are utilized to determine the key slip surface. The following formula can be used to compute the intersection coordinate of the fracture and cross-section:

$$\begin{cases} A(x - x_c) + B(y - y_c) + C(z - z_c) = 0 \\ (x - x_c)^2 + (y - y_c)^2 + (z - z_c)^2 \leq D^2/4 \\ x = x_u \\ A = \cos\alpha\sin\beta \\ B = \sin\alpha\sin\beta \\ C = \cos\beta \end{cases} \tag{14}$$

where $(x_c, y_c, z_c)$ is the coordinate for the center of the fracture disc. The fracture disc's unit normal vector is denoted by $[A, B, C]$. $x_u$ is the coordinate value of the *x*-axis corresponding to the cross-section. The fracture disc's direction is given by $\alpha$. The fracture disc's dip angle is denoted by $\beta$.

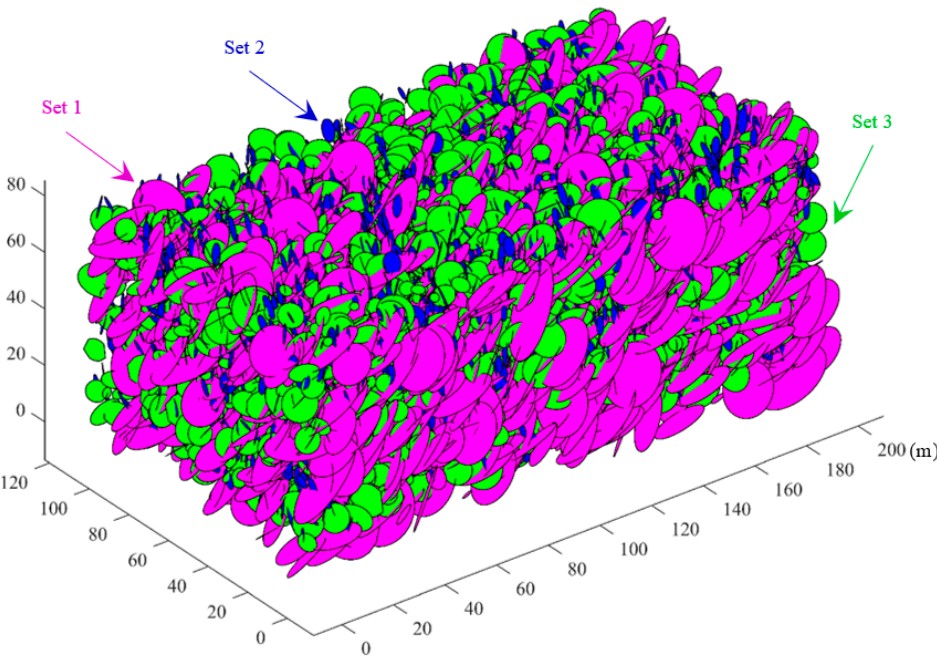

**Figure 6.** 3D fracture network model generated using the Monte Carlo method.

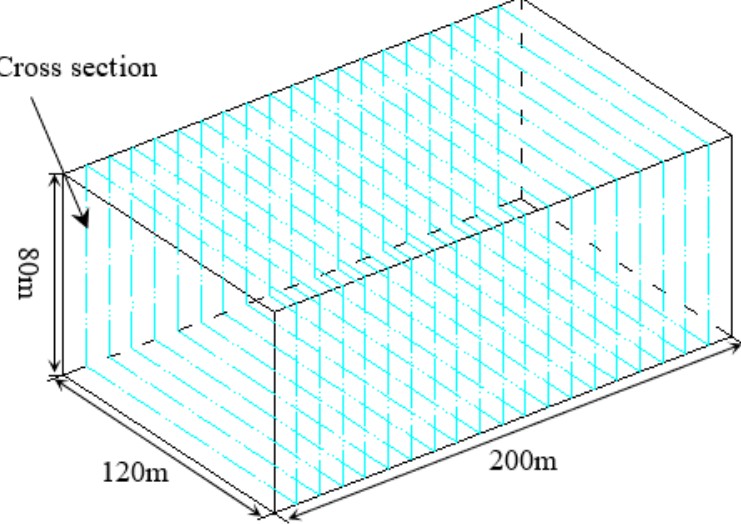

**Figure 7.** Arrangement of the cross sections in the 3D fracture network model.

When the angle between the fracture and cross-section is minimal, the fracture mostly serves as the separating surface of the slope body and has little influence on the slope's sliding. As a result, while looking for the slip surface on a 2D scale, these cracks should be ignored. In this study, fractures are removed when the angles between the fractures and cross-section are less than $30°$.

*2.4. Floyd Algorithm for Searching the Shortest Path*

Rock slopes are characterized by the presence of persistent joints that can cause slope instability [31]. Therefore, it is essential to investigate the spatial distribution of joints. Usually, the shear strength of the fracture is obviously smaller than that of the rock bridge [32]. Thus, the rock slope tends to slide along the joints. In other words, the distribution of fractures determines the critical slip surface of a rock slope. To find the position of the crucial slip surface, a new approach for searching for the shortest path of fracture connectivity is presented based on the aforementioned concepts.

The Floyd algorithm, also known as the insertion point technique, is a dynamic programming method that finds the shortest path between any multisource locations [33]. The shear entrance and exit of the possible slip surface within the rock slope are distinct and fixed. When the potential slip path is shortest, the slip surface goes via the cracks as much as possible while the length of traversing the rock bridges is extremely low. In this case, the shear strength of the sliding surface is naturally the smallest. The Floyd method is used to find the shortest path between the selected shear entrance and exit based on the criteria stated above.

The Floyd algorithm needs to introduce two matrices to store the distance and path between any two points. The element $d_{ij}$ in matrix $D$ represents the distance value between the $i$th point and $j$th point. The element $p_{ij}$ in matrix $P$ represents the point number corresponding to the distance $d_{ij}$ passed from the $i$th point to $j$th point [34]. The weighted adjacency matrix $D^{(0)}$, which is used to store the distance values between any two points is first calculated. All the possible paths through one transfer from the $i$th point to the $j$th point are compared and the shortest path between the $i$th point and the $j$th point arriving directly or passing through only one intermediate point is then selected. The matrix $D^{(1)}$ instead of the matrix $D^{(0)}$ is used to store the shortest paths between any two points. Based on the above principles, the matrices $D^{(2)}, D^{(3)}, \ldots, D^{(k)}$ are calculated in turn. When matrix $D^{(k)}$ and matrix $D^{(k-1)}$ are exactly the same, the iteration terminates. The shortest path information between all point pairs is stored in matrix $D^{(k)}$ and matrix $P^{(k-1)}$. The specific steps of the Floyd algorithm are as follows [35]:

Step 1: Setting the initial distance matrix $D^{(0)} = (d_{ij}^{(0)})$, the matrix $D^{(0)}$ is calculated as:

$$d_{ij}^{(0)} = \begin{cases} W_{ij}, \text{when the } i\text{th point is adjacent to the } j\text{th point} \\ \infty, \text{when the } i\text{th point is not adjacent to the } j\text{th point} \end{cases} \tag{15}$$

where $W_{ij}$ is the calculated distance between the $i$th point and $j$th point according to a certain rule.

Step 2: Constructing the iterative matrix $D^{(k)} = (d_{ij}^{(k)})$, the matrix $D^{(k)}$ is calculated as:

$$d_{ij}^{(k)} = \min\left\{ d_{ir}^{(k-1)} + d_{rj}^{(k-1)} | r = 1, 2, \cdots, n \right\}. \tag{16}$$

Step 3: If $D^{(k)} = D^{(k-1)}$, the iteration terminates. Otherwise, return to Step 2.

## 3. Results and Discussion

When the shear entrance and exit points are provided, the critical slip surface is the shortest path found using the Floyd method. In general, the location of the shear entrance and exit is uncertain. As a result, alternative shear entries and exits are considered to find the shortest path. Finally, the critical slip surface is determined by identifying and calculating the shortest path between multiple shear entrances and exits on 19 cross-sections.

Generally, the shear exits and entrances are arranged around the deformation position at the front and back edge of the slope. In this study, the coordinate points (0, 80), (10, 80), (20, 80), (30, 80), (40, 80), (50, 80), (60, 80), and (70, 80) are separately chosen as the candidate shear entrances and the coordinate points (120, 0) and (120, 10) are separately chosen as the candidate shear exits (Figure 8). With an arbitrary combination of a shear entrance and shear exit, 16 search combinations of the entrance and exit can be formed. Using the Floyd algorithm for searching the shortest path, 16 of the shortest paths are determined. Taking the cross-section of $x$ = 20 m as an example, the two combinations of the entrance point (0, 80) and the exit point (120, 0), the entrance point (70, 80), and the exit point (120, 10) are separately listed. The shortest paths searched corresponding to the above two combinations are separately shown in Figure 9.

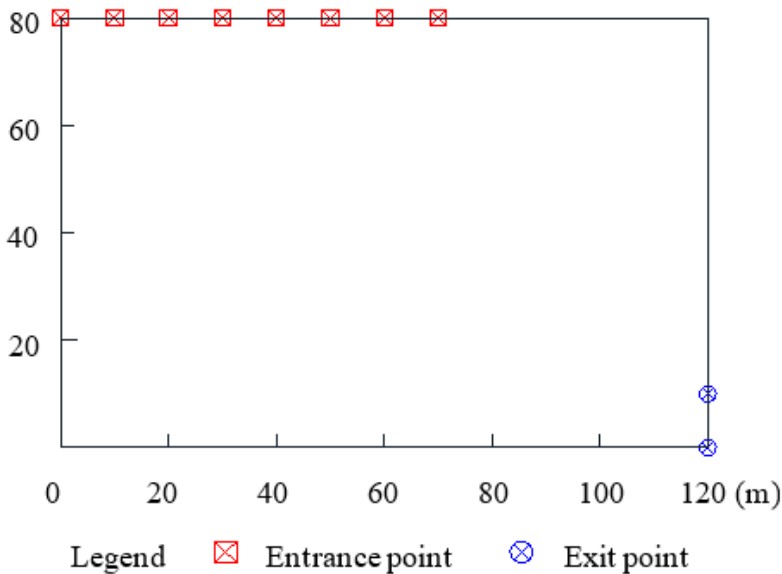

**Figure 8.** Position of the designated entrance point and exit point.

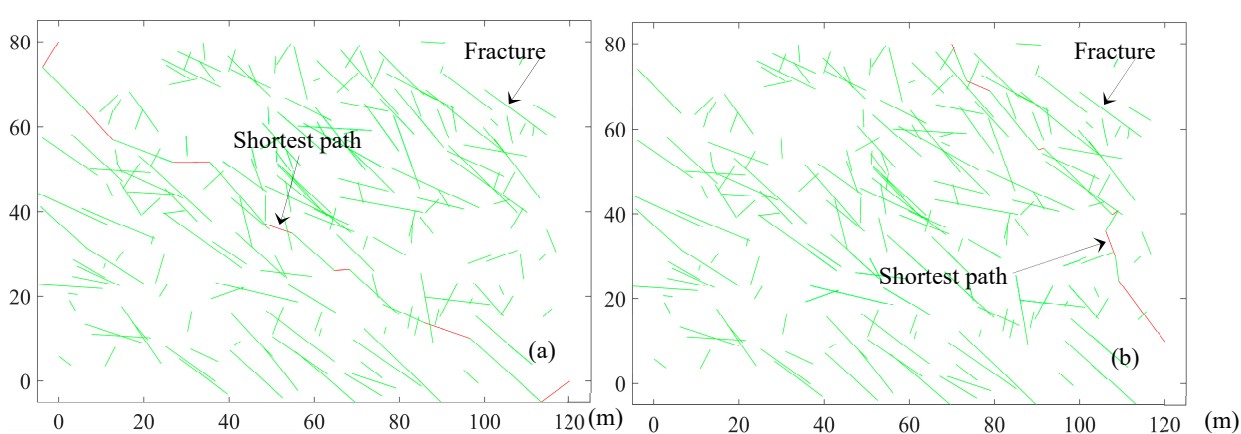

**Figure 9.** The shortest path between the designated shear entrance and exit; (**a**) Entrance point (0, 80) and exit point (120, 0); (**b**) Entrance point (70, 80) and exit point (120, 10).

To date, only the specific information of the fractures exposed on the outcrop can be measured. The fractures developed on the cross-section are usually generated by the idea of probability and statistics [36]. The statistical characteristics of the artificially generated fractures are statistically similar to those of the fractures actually developed within the rock mass. However, the specific location, size, and orientation of the fractures have certain errors. Thus, any one of the shortest path searched by the Floyd algorithm according to

the information of artificial fractures has difficulty representing the actual slip surface. If multiple cross-sections with the fractures are searched using the Floyd algorithm, multiple shortest paths are obtained corresponding to the cross-sections. To a certain extent, the statistical characteristics of multiple shortest paths can be used to reflect the characteristics of the actual slip surface.

In the section "fracture generation in the cross-section of the slope", 19 slope cross-sections with the fractures are generated. Using the Floyd algorithm, the shortest path corresponding to each combination of the entrance point and exit point is searched for in each cross-section. For anyone combination of entrance point and exit point, 19 of the shortest paths can be determined by searching 19 slope cross-sections, as shown in Figure 10. Although the 19 shortest paths are more cluttered, the shear entrance and exit of the shortest paths are basically the same as the shear entrance and exit originally designated. For the sake of safety and convenience, these shortest paths are classified and analyzed using the circular arc that cuts through the shear entrance and exit. It is worth mentioning that the actual slip surface of the slope is not a circular arc, but a linear or fold linear surface. However, the circular arc not only contains the different shortest paths in a statistical sense, it also extends into the interior of the rock slope to the greatest degree. Whether the rock slope is analyzed or supported, it is safe to adopt the circular arc for analysis. According to the above analysis, a circular arc containing different short paths is determined and considered as a potential slip surface. The circular centre and radius of the potential slip surface corresponding to each combination of the entrance point and exit point are calculated and shown in Table 3 and Figure 11.

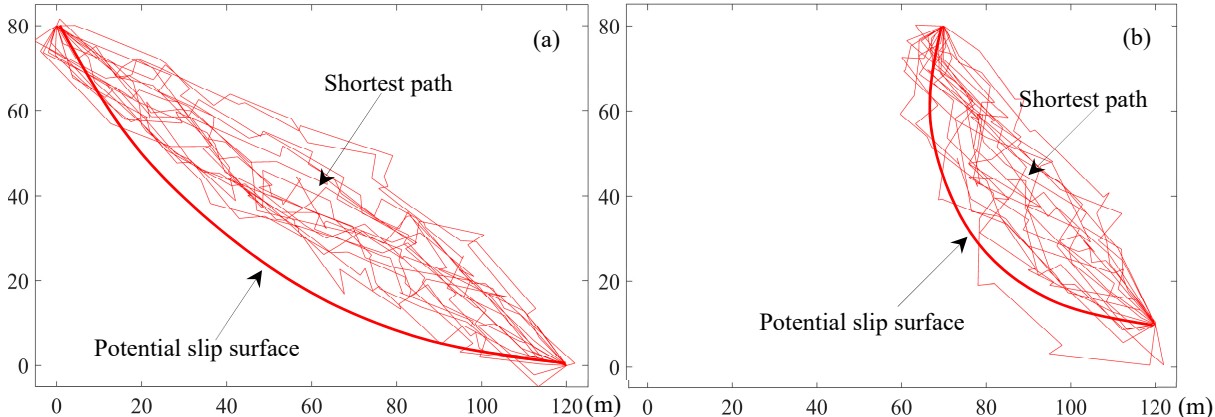

**Figure 10.** The potential slip surface between the designated shear entrance and exit; (**a**) Entrance point (0, 80) and exit point (120, 0); (**b**) Entrance point (70, 80) and exit point (120, 10).

**Table 3.** 16 potential slip surfaces for all combinations of shear entrance and exit.

| Shear Entrance | Shear Exit | Circle Center | Radius | Fracture Frequency | Shear Entrance | Shear Exit | Circle Center | Radius | Fracture Frequency |
|---|---|---|---|---|---|---|---|---|---|
| (0, 80) | | (123.8, 135.6) | 135.7 | 0.742 | (0, 80) | | (109.1, 129.1) | 119.6 | 0.738 |
| (10, 80) | | (125.1, 122.6) | 122.7 | 0.739 | (10, 80) | | (110.5, 116.6) | 107.0 | 0.737 |
| (20, 80) | | (126.1, 110.2) | 110.3 | 0.735 | (20, 80) | | (113.8, 107.5) | 97.7 | 0.731 |
| (30, 80) | (120, 0) | (126.4, 97.9) | 98.1 | 0.734 | (30, 80) | (120, 10) | (114.5, 95.8) | 86.0 | 0.735 |
| (40, 80) | | (127.0, 87.0) | 87.3 | 0.736 | (40, 80) | | (115.8, 85.9) | 76.0 | 0.732 |
| (50, 80) | | (127.3, 77.0) | 77.4 | 0.724 | (50, 80) | | (116.8, 76.8) | 66.9 | 0.720 |
| (60, 80) | | (127.5, 68.1) | 68.5 | 0.715 | (60, 80) | | (117.7, 68.8) | 58.8 | 0.714 |
| (70, 80) | | (127.7, 60.4) | 60.9 | 0.701 | (70, 80) | | (119.6, 62.6) | 52.6 | 0.700 |

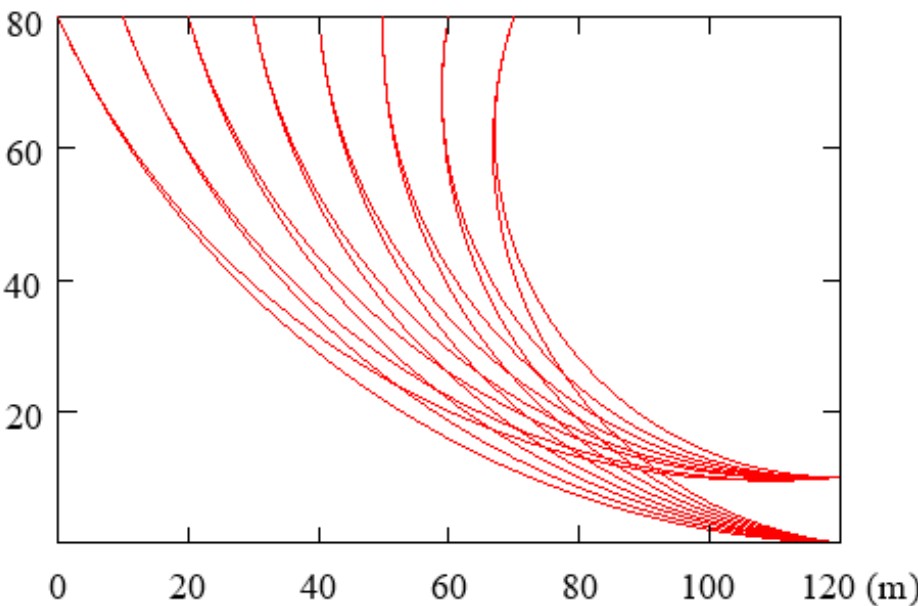

**Figure 11.** A total of 16 potential slip surfaces for the designated shear entrance and exit.

To determine the critical slip surface, the fracture frequency is used to assess the possibility of sliding on a potential slip surface. The fracture frequency is the length ratio of the fractures searched for on the shortest path to the shortest path. The larger the fracture frequency, the longer the length of the fractures searched on the shortest path. The larger the ratio, the easier it will be for the rock bridge between the fractures to break and lead to transfixion, and the easier it will be for the rock slope to fail along the shortest path. The fracture frequency corresponding to each shortest path is calculated by the following formula:

$$f = \frac{\sum\limits_{i=1}^{k} l_i}{L} \tag{17}$$

where $f$ is the fracture frequency. $l_i$ represents the trace length of the $i$th fracture on the shortest path. $k$ indicates the number of fractures searched on the shortest path. $L$ expresses the length of the shortest path.

For any one combination of an entrance point and exit point, the fracture frequency of the shortest path corresponding to each cross-section is calculated. The average of these fracture frequencies is regarded as the fracture frequency of the potential slip surface determined by these shortest paths. The fracture frequency of the possible slip surface through the entrance point (0, 80) and exit point (120, 0), as shown in Table 3, is the largest, with a value of 0.742. When a fractured rock slope fails, the possible slip surface with the largest fracture frequency is frequently chosen because the fractures are the most likely to widen and create the ultimate critical slip surface. The prospective slip surface through the entrance point (0, 80) and exit point (120, 0) might be utilized as the slope's ultimate essential slip surface for safety reasons.

## 4. Conclusions

The discontinuities have a far lower shear strength than rock blocks. Along with the discontinuities, the slip surface is generally simpler to expand. Based on this concept, 222 fractures are quickly obtained by digital close-range photogrammetry, and a new Floyd algorithm is proposed to determine the crucial slip surface of the complex rock slope by locating the shortest path between the fractures. Through the application research of a fracture rock slope in Hengqin Island, Zhuhai, the following discoveries are obtained:

1. Digital close-range photogrammetry technology can quickly establish the 3D model of the slope, which provided a reliable means to obtain information regarding a large number of fracture rock slope fractures quickly and accurately. A 3D fracture network model can be created by interpreting the features of fracture generated on the exposed rock faces. The fracture traces in the cross-section can be obtained by calculating the intersection of the fractures and cross-section.

2. For any one combination of the entrance point and exit point, the shortest paths corresponding to different cross-sections are different and more cluttered. Only the statistical characteristics of multiple shortest paths can be accurately summarized as the characteristics of the actual slip surface. These shortest paths are simplified as a circular arc that contains all shortest paths and is considered as a potential slip surface.

3. For each combination of the entrance point and exit point, a potential slip surface of a circular arc is obtained. The fracture frequency is used to determine the likelihood of sliding along the probable slip surface. Due to the highest fracture frequency, the possible slip surface via the entrance point (0, 80) and exit point (120, 0) is recognized as the slope's final critical slip surface.

**Author Contributions:** Conceptualization, visualization, writing—original draft preparation, and project administration, S.S.; validation, and writing—review and editing, M.Z.; review and formal analysis, C.Z.; methodology, and data curation, F.W.; investigation, and data curation, H.L.; software, resources, M.M.; review, supervision, funding acquisition, and writing—review and editing, C.C. All authors have read and agreed to the published version of the manuscript.

**Funding:** This research was funded by the National Natural Science of China (Grant Nos. 42177139, 42077242, 52104125), Opening Fund of State Key Laboratory of Geohazard Prevention and Geoenvironment Protection (Chengdu University of Technology) (Grant No. SKLGP2018K017), the Natural Science Basic Research Plan in Shaanxi Province of China (Grant No. 2022JQ-304), and the Interdisciplinary Integration and Innovation Project of Jilin University (JLUXKJC2021ZZ17).

**Institutional Review Board Statement:** Not Applicable.

**Informed Consent Statement:** Not Applicable.

**Data Availability Statement:** Not Applicable.

**Acknowledgments:** Thanks to anonymous reviewers and editors for their valuable feedback on the manuscript.

**Conflicts of Interest:** The authors declare no conflict of interest.

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
