# Peer review of "Identification of the Potential Critical Slip Surface for Fractured Rock Slope Using the Floyd Algorithm"

_remotesensing, doi:10.3390/rs14051284_

Round 1

Reviewer 1 Report

see the attachment.  

Just one important suggestions: add some references in the introduction, mainly in the general part where they are few.

Good luck

Author Response

Dear reviewer,

    Thank you very much for your comments on our manuscript entitled “Identification of the potential critical slip surface for the fractured rock slope using the Floyd algorithm” (remotesensing-1592551). Those comments are all valuable and very helpful for revising and improving our manuscript. We have studied comments carefully and have made corrections which we hope meet with approval. Revised portions are marked in red in the revised manuscript. Please see the attachment for major corrections and responses to this paper. Please feel free to contact me, if any further changes are required. We look forward to hearing from you in due course.

    Thank you again!

Sincerely yours,

Chen Cao, Ph.D. & Chun Zhu, Ph.D.

College of Construction Engineering, Jilin University

938 Ximinzhu Road, Changchun 130026, China

Phone number: +86 15948049904

E-mail: ccao@jlu.edu.cn; zhu.chun@hhu.edu.cn

Reviewer 2 Report

Dear authors, manuscript ‘Identification of the potential critical slip surface for the fractured rock slope using the Floyd algorithm’, Manuscript ID: remotesensing-1592551, have some weakness that must be clarified. Please find below some comments:

  1. Generally, the “introduction’ section seems to be appropriate, some state-of-the-art in the field was presented, some lacks mentioned. However, a reader must ‘read between the lines’ to find a piece of main motivation. Both, the novelty and motivation, should be highlighted to make some preview understanding of the studies provided. Please try to put also a more critical review, this would let improve mentioned issues as well.
  2. According to the sentence from lines 84-85 (‘The height of slope 1 is the largest and the angle of the slope surface is the steepest. As a result, slope 1 is chosen as the paper's principal study object (Figure 2).’), a selection of the slope 1 as the biggest value has the influence on the study results. In other words, if the value of each of the slopes was ordered (classified) randomly (or not?), it should be mentioned, if its influence on the results presented. If it was ordered randomly, it must be mentioned as well.
  3. It is not clear if a ‘mathematical model of close-range photogrammetry, line 125, is proposed newly by the author or based on previous studies and, respectively, papers published. This must be unified for each of the equations that if it is based on previous research that should be referenced, even in the beginning of the formulas presence. This can also be decisive in the visibility of the novelty that can be much easier highlighted, if any. Moreover, considering the VirtuoZo system (line 134), the visibility of the novelty is reduced as well.
  4. When analysing both an areal (3D) and cross-section (2D) data, some justification for each type could be introduced as well. Looking for a ‘series of cross-sections’, line 231, it could be also mentioned how the set of data (cross-section) was selected? If randomly, the number of the cross-section can be crucial, otherwise by the selection process, it should be presented in detail. If ‘These cross-sections are perpendicular to the x-axis and arranged at x = 10 m, 20 m, ..., 190 m...’, lines 232-233, and ‘…19 cross-sections with a 2D fracture network are utilized to determine the key slip surface.’, lines 235-236, the selection must be precise consciously.
  5. The proposed method, named a Floyd algorithm, proposed for searching of the shorter path, was not presented according to the data computing time. Even a highly precise approach is presented, the cost in data analysing (computing) would classify each of the procedures as unsuitable. Please try, even approximately, write anything about data computing (time, costs), if any.

Generally, the proposed manuscript in some parts must be significantly improved. Please try to modify the paper according to the above suggestion to make it more suitable for further consideration to Remote Sensing journal publishing.

Author Response

(The authors gave the same response as above.)

Reviewer 3 Report

The authors have presented a method to identify the potential critical slip surface of jointed rock slopes using a fracture network model and digital close-range photogrammetry. The subject is relevant to the scope of the journal and the paper is well organized. The work is original and there is a significant amount of new work in the paper. The paper is recommended for publication after addressing the following comment:

- It is recommended that the authors present 3D potential failure (slip) surfaces, as well (Figure 10).

- In the conventional methods for the rock slope stability analysis, the potential failure (slip) surface is obtained using joint strength parameters (i.e., joint friction angle and cohesion) and unit weight of the rock by calculating the minimum factor of safety. Please clarify how the joint strength is considered in the presented method since there are joints with lower strength in rock slopes that govern the shape and location of the potential failure surface, and the slope failure (slip) does not necessarily occur on the shortest path.

Author Response

(The authors gave the same response as above.)

Round 2

Reviewer 2 Report

Dear Authors of the manuscript ‘Identification of the potential critical slip surface for the fractured rock slope using the Floyd algorithm’, Manuscript ID: remotesensing-1592551.

Firstly, thank you for taking into consideration all of previously the suggested remarks. All of the responses were full, explaining in detail all of the doubts found.

Secondly, very accurate and broadly explanatory answers were located very satisfactory for a regular reader. Such precise and comprehensive answers do not happen very often, unfortunately.

Therefore, according to the review of the revised manuscript, it was found significantly improved and, respectively, can be further considered for publication in the Remote Sensing journal.

Please proceed your studies with further papers as it was found particularly interesting.

Good luck in another studies in future.